# Efficacy of Botulinum Toxin Type-A I in the Improvement of Mandibular Motion and Muscle Sensibility in Myofascial Pain TMD Subjects: A Randomized Controlled Trial

**DOI:** 10.3390/toxins14070441

**Published:** 2022-06-29

**Authors:** Giancarlo De la Torre Canales, Rodrigo Lorenzi Poluha, Natalia Alvarez Pinzón, Bruno Rodrigues Da Silva, Andre Mariz Almeida, Malin Ernberg, Ana Cristina Manso, Leonardo Rigoldi Bonjardim, Célia Marisa Rizzatti-Barbosa

**Affiliations:** 1Clinical Research Unit (CRU), Centro de Investigação Interdisciplinar Egas Moniz (CiiEM), Egas Moniz—Cooperativa de Ensino Superior, CRL, Quinta da Granja, Monte de Caparica, 2829-511 Caparica, Portugal; andremarizalmeida@gmail.com (A.M.A.); mansocristina@gmail.com (A.C.M.); 2Ingá University Center, Uningá, Maringa 87020-900, Brazil; rizzatti@unicamp.br; 3Department of Dentistry, State University of Maringa, Maringa 87020-900, Brazil; rodrigopoluha@gmail.com; 4Institución Universitaria Colegios de Colombia-Centro de Investigación del Colegio Odontológico (CICO) 20, Bogotá 111611, Colombia; natalialvarezodont@gmail.com; 5Private Practice—Boavista Avenue, 4100-139 Porto, Portugal; brunomrsb@hotmail.com; 6Department of Dental Medicine, Karolinska Institutet, and the Scandinavian Center for Orofacial Neurosciences (SCON), 141 52 Huddinge, Sweden; malin.ernberg@ki.se; 7Bauru Orofacial Pain Group, Department of Biological Sciences, Bauru School of Dentistry, University of São Paulo, Sao Paulo 17012-900, Brazil; lbonjardim@fob.usp.br

**Keywords:** botulinum toxin type A, myofascial pain, temporomandibular disorders

## Abstract

This study assessed the effects of botulinum toxin type A (BoNT-A) in mandibular range of motion and muscle tenderness to palpation in persistent myofascial pain (MFP) patients (ReBEC RBR-2d4vvv). Eighty consecutive female subjects with persistent MFP, were randomly divided into four groups (*n* = 20): three BoNT-A groups with different doses and a saline solution group (placebo control group). Treatments were injected bilaterally in the masseter and anterior temporalis muscle in a single session. Clinical measurements of mandibular movements included: pain-free opening, maximum unassisted and assisted opening, and right and left lateral excursions. Palpation tests were performed bilaterally in the masseter and temporalis muscle. Follow-up occurred 28 and 180 days after treatment. For the statistical analysis the Mann–Whitney U-test with Bonferroni correction was used for groups comparisons. Regardless of dose, all parameters of mandibular range of motion significantly improved after 180 days in all BoNT-A groups, compared with the control group. Palpation pain over the masseter and temporalis muscles were significantly reduced in all BoNT-A groups regardless of dose, compared with the control group, after 28 and 180 days of treatment. Independent of doses, BoNT-A improved mandibular range of motion and muscle tenderness to palpation in persistent MFP patients.

## 1. Introduction

Temporomandibular disorders (TMDs) are a heterogeneous group of conditions that affect the masticatory muscles, temporomandibular joint, and associated structures [1]. Myofascial pain (MFP) is one of the most observed TMDs diagnoses, affecting 38% to 75% of the Caucasian population and around 30% of the Asiatic population, and represents 45% of subject with TMD [2]. This condition is defined as a regional muscle pain associated with tenderness to palpation [3] and is most prevalent in young and middle-aged adults (mean age of 30–40 years) and with a higher prevalence in females [2]. The clinical presentation of MFP includes pain referral, tenderness, or pain at the site of palpation, and restriction in the range of motion.

The Research Diagnostic Criteria for TMD (RDC/TMD) [3] and its updated version the Diagnostic Criteria for TMD (DC/TMD) [4] are the most used instruments for TMD diagnoses. They are constituted by a dual-axis system: Axis I, based on the medical history of clinical conditions and Axis II which focuses on pain related disability and psychosocial status assessment [5]. The Axis I protocol is a standardized series of diagnostic tests based on clinical signs and symptoms relying on the outcomes of palpation tests and mandibular range of motion as important findings for diagnostic purposes [3]. Therefore, using the Axis I clinical parameters to assess the outcomes of an MFP treatment in muscle sensitivity and mandibular movements is extremely important in clinical practice and brings reliable data.

Considering the favorable course of TMDs including MFP [6], conservative and reversible treatments (e.g., occlusal splints, behavioral therapies, needling techniques, laser therapy, pharmacotherapy, and physiotherapy) are preferable to more aggressive and irreversible approaches in the initial treatment of MFP since they are effective in reducing painful symptoms [7,8,9,10]. Notwithstanding, the complex etiopathogenesis of MFP [11] make that the use of these conventional and minimally invasive therapies as first line treatment is not enough in some cases [12]. Then, new treatment approaches such as botulinum toxin type A (BoNT-A) are proposed with a promising potential to diminish pain [13]. In addition, since the US-FDA approved BoNT-A as a treatment for many muscle and pain disorders [14], animal studies have showed that BoNT-A have analgesic effects and inhibits nociceptive mediators release (peripherally and centrally), a mechanism separate of its neuromotor effect.

According to the American Academy of Neurology, BoNT-A is classified as a level B treatment (possibly effective) for MFP [15]. However, literature is still not clear about the efficacy of BoNT-A in the improvement of MFP, since well-designed clinical trials presented opposite results when compared BoNT-A with a placebo, and not demonstrated the superiority of BoNT-A over classic treatments like oral appliances [16]. Flaws like the lack of standardized protocols of application and a proper methodological design, contribute to this lack of consensus [16]. Additionally, even though BoNT-A treatment is considered generally safe, a recent systematic review concluded that BoNTA injection could results in mandibular bony change, which may hinder BoNT-A benefits [17].

Based on the conflicting results available on the use of BoNT-A for MFP, it is im-portant to assess treatment effect on other parameters besides pain intensity in patients with MFP. Especially since this toxin may have a use as an alternative in cases where the conventional treatments fail. Thus, the aim of this randomized, placebo controlled clinical trial was to demonstrate the efficacy of BoNT-A in the improvement of mandibular range of motion and muscle sensibility to palpation in persistent MFP patients.

## 2. Results

### 2.1. Subjects

Eighty consecutive female subjects, who were diagnosed with MFP, according to the Brazilian Portuguese version of the RDC/TMD [3] at the TMD Clinic of Piracicaba Dental School, University of Campinas, São Paulo, Brazil, were included. Subjects were assessed by two calibrated researchers not involved in any other process of the study (kappa coefficient = 0.80 for RDC/TMD inter-examiner assessment).

### 2.2. Mandibular Motion

Considering the pain-free mouth opening, there was no difference between groups at baseline and at the 28 days follow-up (*p* > 0.05), and no changes within groups during the first month (*p* > 0.05) of follow-up (Table 1). However, at the 180-day evaluation, all BoNT-A groups showed a significant improvement compared with the other evaluation periods (*p* < 0.05). In contrast, a decrease in pain-free mouth opening was found in the SS group. In the between group comparisons, all BoNT-A groups presented a greater increase (*p* < 0.05) in pain-free mouth opening compared with the control group at the 180-day evaluation (Table 1).

The same pattern was observed for maximum unassisted and assisted mouth opening (Table 2). A significant improvement was found at the 180-day follow-up only in the BoNT-A groups. Moreover, the groups showed higher unassisted and assisted mouth opening in the BoNT-A groups compared with SS (*p* < 0.05) at the last follow-up (Table 2).

Regarding right and left lateral movements, no significant differences were found in the group comparisons at baseline and after 28 days of treatment (*p* < 0.05). However, in the 180-day evaluation the BoNT-A groups showed a significant increase in all lateral movements (*p* < 0.05) without significant differences among them. In contrast, no significant differences were found in the SS group in any period of evaluation. Furthermore, BoNT-A groups showed higher values at the 180-day follow-up (*p* < 0.05) compared with the SS group (Table 3).

### 2.3. Muscle Pain

Considering the mean of pain on palpation in masseter and temporalis muscles of both sides (Table 4 and Table 5), the BoNT-A groups reported significantly reduced pain sensbility to palpation at the 28-day follow-up (*p* < 0.05). This significant improvement remained until the 180-day evaluation (*p* < 0.05), but without differences between BoNT-A groups (*p* > 0.05). The SS group showed no significant differences in any period of assessment (*p* > 0.05). Comparisons between BoNT-A and SS showed lower values for BoNT-A groups at the 28- and 180-day follow-ups (*p* < 0.05).

## 3. Discussion

To the best of our knowledge, this is the first randomized, placebo-controlled clinical trial to demonstrate that independent of dosage, BoNT-A improves mandibular range of motion (pain-free opening of the mouth, maximum unassisted and assisted mouth opening, and right and left lateral mandibular movements) and muscle sensibility to palpation (in masseter and temporal muscles bilateral) in persistent MFP patients compared with a placebo.

Restriction in mandibular range of motion is one of the most common clinical presentations of MFP and usually it is used as a parameter to assess treatment efficacy. One of the most relevant results from the present study was the improvement of mandibular range of motion in persistent MFP patients treated with BoNT-A (Table 1, Table 2 and Table 3). After 180 days of treatment, all BoNT-A groups showed a significant increase in pain-free mouth opening, maximum unassisted and assisted mouth opening, and right and left lateral mandibular movements. These results corroborate available literature that also presented a gain of mandibular range of motion in MFP patients when treated with BoNT-A (100 to 150 U, distributed among masseter, temporal, and medial pterygoid muscle bilateral) compared with injections of saline solution and lidocaine [8,18]. However, since other studies such as the one performed by Ernberg et al., 2011 [19] reported no significant changes for these parameters, no definitive conclusions can be made as to whether BoNT-A injections improve mandibular range of motion. It is interesting to note that in the present study, the increase in mandibular range of motion was not influenced by the BoNT-A doses; all groups treated with BoNT-A had a greater increase compared with the control group. In fact, patients in the control group, after 180 days of follow-up, showed decreased values of pain-free mouth opening as well as unassisted and assisted mouth opening, demonstrating that SS was not effective in ameliorating these parameters.

In the research field and in every day clinical practice it is more and more evident that BoNT-A has an antinociceptive effect in muscle conditions [13,14,16]. Considering the results of our study regarding pain sensibility on palpation in masseter and temporalis muscles, BoNT-A groups showed significantly reduced pain sensitivity to palpation 28 days after treatment and this improvement remained for 180 days after treatment. The improvement in this pain parameter is in accordance with other studies where patients with persistent MFP treated with BoNT-A, presented a significant reduction in different pain measurements including the pain pressure threshold of the masticatory muscles [20,21,22,23]. Moreover, the fact that in our study, the positive effects of a single injection of BoNT-A in the assessed variables lasted for 6 months, can clinically explain that the antinociceptive effect of BoNT-A is independent of muscle paralysis which last for 3 to 4 months [24].

Taken together, the increased mandibular range of motion and the reduction in muscle sensibility to palpation found in the present study after the BoNT-A treatment, warrants some discussion. First, the improvement of all variables related to the mandibular range of motion was significant only 180 days after treatment, while a significant improvement in the muscle sensibility to palpation was observed already 28 days after treatment. These findings are understandable since it is well-known that pain can restrict movements, whether by kinesiophobia and/or muscle protective contraction [25,26,27,28]. Therefore, it is expected that the mandibular range of motion only improves after a significant reduction of muscle sensibility. If this study had a monthly follow-up of the variables, this relationship would be better elucidated. Additionally, this reasoning also serves as an alert for the importance of pain improvement as soon as possible in cases of MFP in order to obtain a quicklier improvement in mandibular movements or a reduction in movement pain. Second, even though in our study BoNT-A was only injected in the masseter and temporalis muscles, a significant improvement in the assessed mandibular movements in which other muscles such as medial and lateral pterygoid participate was also achieved. These results demonstrate that by diminishing muscle pain, BoNT-A can perhaps reestablish the equilibrium in masticatory musculature function [29,30].

Some animal and clinical studies have demonstrated that BoNT-A injections in masticatory muscles could produce adverse effects such as: alter masticatory performance, in-duce muscle atrophy, and diminish mandibular bone volume [31]. In addition, these ad-verse effects are proportional with the doses and number of applications of BoNT-A [31]. Interestingly, the positive results found in the BoNT-A groups in our study, were not ifluenced by the BoNT-A doses, showing that low doses were equally efficient as median and higher doses in improving mandibular range of motion and muscle palpation pain in patients with persistent MFP. These results are in line with our previous study comparing BoNT-A doses [32] in which pain intensity and sensibility (pain pressure threshold) were improved regardless of dosage. In addition, no adverse effects in muscle and bone tissue with low doses of BoNT-A were found in that study. Notwithstanding, in our previous study [32] a single injection of 50U and 70U and 20U and 25U in masseter and anterior temporalis muscles respectively, produced a reduction in muscles thickness and a de-crease in mandibular bone volume (coronoid process and mandibular head), showing that adverse effects are doses dependent [32]. Therefore, it could be proposed that regardless of doses, BoNT-A also diminish pain and improve mandibular movements [23] with low risks of developing serious adverse effects when low doses are used.

Nevertheless, caution is suggested when judging the present findings, since this investigation was performed on a restricted population (persistent MFP) and without gender comparison. Future studies with larger samples, using even lower doses of BoNT-A, com-paring its efficacy with other treatments are encouraged. Additionally, due to BoNT-A potential to develop adverse effects, we recommend indicating this treatment just in myofascial pain TMD patients who did not obtain a substantial pain relieve with conservative treatments.

## 4. Conclusions

In view of the results and limitations of this study, it can be concluded that BoNT-A, independent of dosage, improves mandibular range of motion (pain-free opening of the mouth, maximum unassisted and assisted mouth opening, and right and left lateral mandibular movements) and muscle pain to palpation of the masseter and temporal muscles in persistent MFP patients compared with SS injections.

## 5. Methods

The present study was approved by the Research Ethics Committee of Piracicaba Dental School (CAAE # 22953113.8.0000.5418-Date: 4 February 2014) and the Brazilian Regis-try of Clinical Trials (ReBEC RBR-2d4vvv). Subjects were recruited at the TMD Clinic of Piracicaba Dental School, University of Campinas, São Paulo, Brazil, from June 2014 to May 2017. All subjects provided a written informed consent to participate in this clinical trial This study was based on a secondary data analysis of our previous publication De la Tor-re Canales et al., 2020 [29].

### 5.1. Inclusion Criteria

Years of age: 18–45;A diagnosis of MFP according to the RDC/TMD classification [3]No significant pain relief after at least three months of previous treatment;A baseline pain intensity of at least 50 on a visual analogue scale (VAS: 0–100);Subjects were also included if they presented other non-painful TMD diagnoses besides MFP.

### 5.2. Exclusion Criteria

History of trauma in the face and neck area;Dental pain in the last 6 months;Systemic diseases (arthritis and arthrosis) and major psychiatric disorders;Current use of drugs acting on neuromuscular junctions;Contraindication or hypersensitivity to BoNT-A;Received anti-tetanus vaccine in the 3 months before the start of the clinical trial.

The sample was randomly divided into four equal groups (*n* = 20): BoNT-A low (BoNTA-L/10 U in each temporalis and 30 U in each masseter), BoNT-A medium (BoNTA-M/20 U in each temporalis and 50 U in each masseter), BoNT-A high (BoNTA-H/25 U in each temporalis and 75 U in each masseter), and saline solution 0.9% ((SS) placebo control group). To blind applications, each muscle received an injection containing 1 mL of the correspondent reconstituted BoNT-A-L/M/H doses or SS by a clinician not involved in the reconstitution process. Randomization was performed using a computer software (Isfahan, Iran https://random-allocation-software.software.informer.com/2.0/ (accessed on 15 November 2013) operated by a technician not involved in any other procedures in the study.

### 5.3. Treatments

A counseling session was performed for all the included subjects at the first appointment by a single trained researcher. In brief, counseling consisted in educating the subjects about the anatomic characteristics and physiology of the stomatognathic system and the etiology, evolution, and prognosis of MFP, teaching self-care strategies for pain and to control parafunctional habits and giving information about improvement of sleep and the importance of dietary habits.

BoNT-A (100 U; Botox^®^, Allergan, Irvine, CA, USA) was reconstituted using non-preserved sterile saline solution 0.9%. Doses of BoNT-A were based on a previous report [20,29]. Bilateral intramuscular injections were performed in the masseter and anterior temporalis muscles using a 1-mL syringe with a 30-gauge needle by an investigator who was not involved in the dilution process. Subjects were asked to first clench their teeth to delimit the muscle area (masseter and anterior temporalis) and then relax the muscle whereafter a total of five injections per muscle with a separation of 5 mm between them were applied. Injections of BoNT-A or SS were performed during a single appointment. Participants and investigators assessing the outcomes were masked to all treatment assignments.

### 5.4. RDC/TMD Axis I Assessment

Mandibular movements and muscle sensitivity to palpation were evaluated in accordance with RDC/TMD-Axis I [3]. The RDC/TMD Axis I includes a clinical examination of range of mandibular movements and pain upon movement, assessment of joint sounds, and temporomandibular joint and muscle palpation. A diagnosis of MPF is based on pain-free opening capacity (mm) and number of muscle sites painful to palpation. Participants were evaluated while seated in a dental chair, in a room with adequate lighting. Clinical measurements of the mandibular movements were made by using a digital caliper rule (Mitutoyo 500- 144B—Suzano, São Paulo, Brazil), and included: pain-free opening, maximum unassisted and assisted opening, and right and left lateral movements. Palpation tests were performed bilaterally to six different areas of the face and head (origin, body, and insertion of masseter muscle and posterior, middle, and anterior temporalis muscle; the arithmetic mean of the three evaluation points in each muscle was considered for statistical analysis). Subjects’ pain while palpation was recorded on a 4-point ordinal scale: 0 = no pain, 1 = mild pain, 2 = moderate pain, and 3 = severe pain.

### 5.5. Data Evaluation and Statistical Analyses

Data from mandibular movements and muscle tenderness to palpation were assessed at three time points: before, and 28 as well as 180 days after treatment. All variables’ results were expressed as median, minimum, maximum, and means ± standard deviation (SD) and were assessed for normal distribution with the Shapiro–Wilk test. There was no normal distribution. Then, for non-parametric multiple comparison between groups Mann–Whitney U-test with Bonferroni correction was used. All data were analyzed using SPSS Statistics 25.0 software (IBM^®^, New York, NY, USA). A 5% probability level was considered significant in all tests.

## Figures and Tables

**Table 1 toxins-14-00441-t001:** Mean and standard deviation of pain-free opening (mm) in different evaluation periods.

Time						
	Baseline		28 Days		180 Days	
Groups	Mean	SD	Mean	SD	Mean	SD
BoNTA-L	31.9 aB	8.7	33.6 aB	8.8	38.3 aA	7.5
BoNTA-M	32.4 aB	9.4	35.0 aB	7.7	40.7 aA	7.6
BoNTA-H	32.6 aB	8.9	35.0 aB	9.2	39.0 aA	8.1
SS	36.2 aB	6.1	37.6 aB	6.6	34.0 bA	9.0

Different letters (lowercase in vertical) represent significant difference (*p* ≤ 0.05) among groups. Different letters (uppercase in horizontal) represent significant difference (*p* ≤ 0.05) among evaluation periods. BoNTA: botulinum toxin type A; L: low; M: median; H: high; SS: saline solution; SD: standard deviation.

**Table 2 toxins-14-00441-t002:** Median, minimum, and maximum values for maximum unassisted and assisted opening in different evaluation periods.

	Time									
		Baseline			28 Days			180 Days		
	Groups	Median	Mn	Mx	Median	Mn	Mx	Median	Mn	Mx
Unassisted	BoNTA-L	41.0 aB	28.0	59.0	42.0 aB	29.0	58.0	44.5 aA	31.0	60.0
	BoNTA-M	43.0 aB	29.0	54.0	43.5 aB	30.0	52.0	45.0 aA	35.0	60.0
	BoNTA-H	41.0 aB	26.0	50.0	43.0 aB	25.0	52.0	46.0 aA	23.0	55.0
	SS	41.5 aB	35.0	55.0	42.0 aB	50.0	50.0	39.0 bB	20.0	45.0
Assisted	BoNTA-L	43.0 aB	30.0	60.0	43.0 aB	31.0	58.0	47.0 aA	33.0	62.0
	BoNTA-M	47.0 aB	30.0	57.0	45.5 aB	31.0	55.0	48.5 aA	35.0	61.0
	BoNTA-H	44.0 aB	30.0	55.0	47.0 aB	30.0	53.0	50.0 aA	31.0	57.0
	SS	42.5 aB	33.0	56.0	43.5 aB	34.0	51.0	41.5 bB	22.0	47.0

Different letters (lowercase in vertical) represent significant difference (*p* ≤ 0.05) among groups. Different letters (uppercase in horizontal) represent significant difference (*p* ≤ 0.05) among evaluation periods. BoNTA: botulinum toxin type A; L: low; M: median; H: high; SS: Saline Solution. Mn: minimum; Mx: maximum.

**Table 3 toxins-14-00441-t003:** Median, minimum, and maximum values for right and left lateral movements in different evaluation periods.

	Time									
		Baseline			28 Days			180 Days		
Groups		Median	Mn	Mx	Median	Mn	Mx	Median	Mn	Mx
Right	BoNTA-L	9.5 aA	2.0	15.0	9.0 aA	3.0	12.0	11.0 aB	6.0	13.0
	BoNTA-M	9.0 aA	5.0	12.0	9.0 aA	6.0	14.0	10.0 aB	7.0	13.0
	BoNTA-H	8.5 aA	4.0	14.0	9.5 aA	4.0	14.0	10.5 aB	5.0	15.0
	SS	8.0 aA	3.0	11.0	8.5 aA	5.0	14.0	9.0 bA	5.0	12.0
Left	BoNTA-L	8.0 aA	0.0	13.0	9.0 aA	4.0	13.0	9.5 aB	4.0	14.0
	BoNTA-M	8.0 aA	2.0	12.0	9.0 aA	6.0	15.0	10.0 aB	8.0	15.0
	BoNTA-H	9.5 aA	4.0	12.0	9.5 aA	4.0	15.0	10.0 aB	6.0	15.0
	SS	8.0 aA	3.0	13.0	8.5 aA	5.0	15.0	8.5 bA	5.0	15.0

Different letters (lowercase in vertical) represent significant difference (*p* ≤ 0.05) among groups. Different letters (uppercase in horizontal) represent significant difference (*p* ≤ 0.05) among evaluation periods. BoNTA: botulinum toxin type A; L: low; M: median; H: high; SS: Saline Solution. Mn: minimum; Mx: maximum.

**Table 4 toxins-14-00441-t004:** Median, minimum, and maximum values for right and left masseter muscle in different evaluation periods.

	Time									
		Baseline			28 Days			180 Days		
	Groups	Median	Mn	Mx	Median	Mn	Mx	Median	Mn	Mx
Right	BoNTA-L	2.0 aA	0.0	3.0	1.0 bB	0.0	12.0	0.5 bB	0.0	2.0
	BoNTA-M	1.0 aA	0.0	3.0	0.0 bB	0.0	14.0	0.0 bB	0.0	3.0
	BoNTA-H	2.0 aA	0.0	3.0	0.5 bB	0.0	14.0	0.0 bB	0.0	3.0
	SS	2.0 aA	1.0	3.0	2.0 aA	0.0	14.0	2.0 aA	0.0	3.0
Left	BoNTA-L	1.5 aA	0.0	3.0	0.0 bB	0.0	2.0	0.5 bB	0.0	3.0
	BoNTA-M	1.0 aA	0.0	3.0	0.0 bB	0.0	3.0	0.0 bB	0.0	3.0
	BoNTA-H	2.0 aA	0.0	3.0	0.0 bB	0.0	3.0	0.0 bB	0.0	2.0
	SS	1.0 aA	0.0	3.0	2.0 aA	0.0	3.0	2.0 aA	0.0	3.0

Different letters (lowercase in vertical) represent significant difference (*p* ≤ 0.05) among groups. Different letters (uppercase in horizontal) represent significant difference (*p* ≤ 0.05) evaluation periods. BoNTA: botulinum toxin type A; L: low; M: median; H: high; SS: Saline Solution. Mn: minimum; Mx: maximum.

**Table 5 toxins-14-00441-t005:** Median, minimum, and maximum values for right and left temporal muscle in different evaluation periods.

	Time									
		Baseline			28 Days			180 Days		
	Groups	Median	Mn	Mx	Median	Mn	Mx	Median	Mn	Mx
Right	BoNTA-L	2.5 aA	0.0	3.0	1.0 bB	0.0	2.0	0.0 aB	0.0	2.0
	BoNTA-M	2.0 aA	0.0	3.0	1.0 bB	0.0	2.0	0.0 aB	0.0	2.0
	BoNTA-H	2.0 aA	0.0	3.0	0.0 bB	0.0	3.0	0.5 aB	0.0	2.0
	SS	2.0 aA	0.0	3.0	2.0 aA	0.0	3.0	2.0 bA	0.0	3.0
Left	BoNTA-L	2.0 aA	0.0	3.0	1.0 bB	0.0	3.0	0.5 aB	0.0	2.0
	BoNTA-M	1.5 aA	0.0	3.0	0.0 bB	0.0	3.0	0.0 aB	0.0	2.0
	BoNTA-H	2.0 aA	0.0	3.0	0.5 bB	0.0	3.0	0.0 aB	0.0	2.0
	SS	1.5 aA	0.0	3.0	1.0 bA	0.0	3.0	1.5 bA	0.0	3.0

Different letters (lowercase in vertical) represent significant difference (*p* ≤ 0.05) among groups. Different letters (uppercase in horizontal) represent significant difference (*p* ≤ 0.05) evaluation periods. BoNTA: botulinum toxin type A; L: low; M: median; H: high; SS: Saline Solution. Mn: minimum; Mx: maximum.

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
