# Peer review of "Efficacy of Botulinum Toxin Type-A I in the Improvement of Mandibular Motion and Muscle Sensibility in Myofascial Pain TMD Subjects: A Randomized Controlled Trial"

_toxins, 2022, doi:10.3390/toxins14070441_

Round 1
Reviewer 1 Report
The work is devoted to the treatment of temporomandibular disorders, widespread throughout the world, and is a continuation of earlier studies. The number of patients and parameters observed are sufficient to study the therapeutic effect of intramuscular injections of different doses of botulinum neurotoxin A. Reliable results have been obtained. As recommendations to improve the article, I can suggest adding data on the manifestation of side effects when using high doses of BoNT-A and general recommendations on the use of the toxin for the treatment of tromporomandibular pain.
Author Response
We would like to thank the reviewer fot the valuable suggestions. Please, find our point by point responses above.
- I can suggest adding data on the manifestation of side effects when using high doses of BoNT-A.
- Adding general recommendations on the use of the toxin for the treatment of temporomandibular pain.
Reviewer 2 Report
Dear Authors,
The aim of this randomized, placebo controlled clinical trial was to demonstrate if botulinum toxin type A could improve mandibular range of motion and muscle sensibility to palpation in persistent myofascial pain patients.
In my opinion, the manuscript is well written, respecting the author’s guidelines.
The study is of scientific interest and in line with the aims of the journal. The author guidelines have been respected. The Result and Discussion sections were well described.
References were well reported.
However, there are some minor issues that should be addressed.
Minor revisions
Title
Accorging to the study design (RCT), I suggest modifying the title as follow: Efficacy of Botulinum Toxin Type–A in the Improvement of Mandibular Motion and Muscle Sensibility in Myofascial Pain TMD Subjects: A Randomized Controlled Trial
Abstract
Please report the RCT registration number in the abstract.
Main text
- In the text, reference numbers should be placed in square brackets [ ], and placed before the punctuation; for example [1], [1–3] or [1,3]. Follow instruction for authors (https://www.mdpi.com/journal/toxins/instructions).
- Insert page numbers and add line numbers to the manuscript document to aid reviewers and editors.
Introduction
- I suggest improving the Introduction Section, better reporting the most common TMD treatment and approaches. Accordingly, you might cite and discuss the following papers: doi: 10.3233/BMR-210236, doi: 10.5606/tftrd.2021.6615, doi: 10.1111/joor.13326.
- According to the study design (RCT), I suggest rephrasing the aim of the study as follow: “Thus, the aim of this randomized, placebo controlled clinical trial was to demonstrate the efficacy of BoNT-A in the improvement of mandibular range of motion and muscle sensibility to palpation in persistent MFP patients.
Methods
- “Eighty consecutive female subjects, who were diagnosed with MFP, according to the Brazilian Portuguese version of the RDC/TMD 3 at the TMD Clinic of Piracicaba Dental School, University of Campinas, São Paulo, Brazil, were included. Subjects were assessed by two calibrated researchers not involved in any other process of the study (kappa coefficient = 0.80 for RDC/TMD inter-examiner assessment).” Put this information in the Results section.
- Please better describe the setting and the duration of the recruitment.
- Why do you use RDC/TMD and not DC/TMD (Shiffmann et al. 2014)?
Author Response
We would like to thanks the reviewer for all the valuable suggestions, that certainly improved our manuscript. Please find above our point by point answers to your considerations.
- Title: Accorging to the study design (RCT), I suggest modifying the title as follow: Efficacy of Botulinum Toxin Type–A in the Improvement of Mandibular Motion and Muscle Sensibility in Myofascial Pain TMD Subjects: A Randomized Controlled Trial
Our responde: We modified the title according to the reviewer's suggestion (page 1 - lines 1-4).
Revised text: Efficacy of Botulinum Toxin Type–A I in the Improvement of Mandibular Motion and Muscle Sensibility in Myofascial Pain TMD Subjects: A Randomized Controlled Trial
- Abstract: Please report the RCT registration number in the abstract.
Our responde: We added the RCT registration number according to the reviewer's suggestion (page 1 - line 7).
Revised text:This study assessed the effects of botulinum toxin type A (BoNT-A) in mandibular range of motion and muscle tenderness to palpation in persistent myofascial pain (MFP) patients (ReBEC RBR-2d4vvv).
- Main text: In the text, reference numbers should be placed in square brackets [ ], and placed before the punctuation; for example [1], [1–3] or [1,3]. Follow instruction for authors (https://www.mdpi.com/journal/toxins/instructions). Insert page numbers and add line numbers to the manuscript document to aid reviewers and editors.
Our responde: We corrected the format of the citations according to the author's instructions.
Revised text: The hall manuscript.
- Introduction: I suggest improving the Introduction Section, better reporting the most common TMD treatment and approaches. Accordingly, you might cite and discuss the following papers: doi: 10.3233/BMR-210236, doi: 10.5606/tftrd.2021.6615, doi: 10.1111/joor.13326.
Our responde: We appreciate the reviewers suggestions, however, we consider that as the main focus of our manuscript is to report BoNT-A effects and efficacy on MFP, getting deeper in other MFP treatments could make the introduction section left out of context. Notwithstanding, we modified the text to better address this issue (pages 1-2 - lines 45-54).
Revised text: Considering the favorable course of TMDs including MFP [6] conservative and reversible treatments (e.g., occlusal splints, behavioral therapies, needling techniques, laser therapy, pharmacotherapy, and physiotherapy) are preferable to more aggressive and irreversible approaches in the initial treatment of MFP since they are effective in reducing painful symptoms [7,8].
- According to the study design (RCT), I suggest rephrasing the aim of the study as follow: “Thus, the aim of this randomized, placebo controlled clinical trial was to demonstrate the efficacy of BoNT-A in the improvement of mandibular range of motion and muscle sensibility to palpation in persistent MFP patients.
Our responde: We rephrased the aim of the study according to the reviewer's suggestion (page 2 - lines 70-72).
Revised text: Thus, the aim of this randomized, placebo controlled clinical trial was to demonstrate the efficacy of BoNT-A in the improvement of mandibular range of motion and muscle sensibility to palpation in persistent MFP patients.
- Methods: “Eighty consecutive female subjects, who were diagnosed with MFP, according to the Brazilian Portuguese version of the RDC/TMD 3 at the TMD Clinic of Piracicaba Dental School, University of Campinas, São Paulo, Brazil, were included. Subjects were assessed by two calibrated researchers not involved in any other process of the study (kappa coefficient = 0.80 for RDC/TMD inter-examiner assessment).” Put this information in the Results section.
Our responde: We moved this text to the results section (page 2 - lines 78-82).
Revised text: 2.1 Subjects: Eighty consecutive female subjects, who were diagnosed with MFP, according to the Brazilian Portuguese version of the RDC/TMD 3 at the TMD Clinic of Piracicaba Dental School, University of Campinas, São Paulo, Brazil, were included. Subjects were assessed by two calibrated researchers not involved in any other process of the study (kappa coefficient = 0.80 for RDC/TMD inter-examiner assessment).
- Please better describe the setting and the duration of the recruitment.
Our responde: We describe in a better way the duration of the recruitment in the methods section (page 6 - lines 225-231).
Revised text: The present study was approved by the Research Ethics Committee of Piracicaba Dental School (CAAE # 22953113.8.0000.5418-Date: 04/02/2014) and the Brazilian Registry of Clinical Trials (ReBEC RBR-2d4vvv). Subjects were recruited at the TMD Clinic of Piracicaba Dental School, University of Campinas, São Paulo, Brazil, from June 2014 to May 2017. All subjects provided a written informed consent to participate in this clinical trial.
- Why do you use RDC/TMD and not DC/TMD (Shiffmann et al. 2014)?
Our responde: We did not use the DC/TMD, because the Brazilian version of this exam (DC/TMD) was only published in 2019. However, we used the Brazilian version of the RDC/TMD, which is a validated tool for TMD. diagnosis.
Round 2
Reviewer 2 Report
Dear Authors,
You have improved the paper following my revisions. Furthermore, the Introduction Section must be improved, describing the most common TMD treatment and approaches. Accordingly, you should cite and discuss the following papers:
Ferrillo M, Ammendolia A, Paduano S, Calafiore D, Marotta N, Migliario M, Fortunato L, Giudice A, Michelotti A, de Sire A. Efficacy of rehabilitation on reducing pain in muscle-related temporomandibular disorders: A systematic review and meta-analysis of randomized controlled trials. J Back Musculoskelet Rehabil. 2022 Feb 18. doi: 10.3233/BMR-210236.
Deregibus A, Ferrillo M, Grazia Piancino M, Chiara Domini M, de Sire A, Castroflorio T. Are occlusal splints effective in reducing myofascial pain in patients with muscle-related temporomandibular disorders? A randomized-controlled trial. Turk J Phys Med Rehabil. 2021 Mar 4;67(1):32-40. doi: 10.5606/tftrd.2021.6615.
Owen M, Gray B, Hack N, Perez L, Allard RJ, Hawkins JM. Impact of botulinum toxin injection into the masticatory muscles on mandibular bone: A systematic review. J Oral Rehabil. 2022 Jun;49(6):644-653. doi: 10.1111/joor.13326.
Author Response
We thank the reviewer for the suggestions. Find our point by point response below
- Accordingly, you should cite and discuss the following papers:
- Ferrillo M, Ammendolia A, Paduano S, Calafiore D, Marotta N, Migliario M, Fortunato L, Giudice A, Michelotti A, de Sire A. Efficacy of rehabilitation on reducing pain in muscle-related temporomandibular disorders: A systematic review and meta-analysis of randomized controlled trials. J Back Musculoskelet Rehabil. 2022 Feb 18. doi: 10.3233/BMR-210236.
- Deregibus A, Ferrillo M, Grazia Piancino M, Chiara Domini M, de Sire A, Castroflorio T. Are occlusal splints effective in reducing myofascial pain in patients with muscle-related temporomandibular disorders? A randomized-controlled trial. Turk J Phys Med Rehabil. 2021 Mar 4;67(1):32-40. doi: 10.5606/tftrd.2021.6615.
- Owen M, Gray B, Hack N, Perez L, Allard RJ, Hawkins JM. Impact of botulinum toxin injection into the masticatory muscles on mandibular bone: A systematic review. J Oral Rehabil. 2022 Jun;49(6):644-653. doi: 10.1111/joor.13326.
Our response: The text was modified and the suggested references were added, pages 1,2 lines 44-67.
Revised text: Considering the favorable course of TMDs including MFP [6], conservative and reversible treatments (e.g., occlusal splints, behavioral therapies, needling techniques, laser therapy, pharmacotherapy, and physiotherapy) are preferable to more aggressive and irreversible approaches in the initial treatment of MFP since they are effective in reducing painful symptoms [7-10].
Our response: The text was modified and the suggested references were added, pages 2 lines 81-83.
Revised text: Additionally, even though BoNT-A treatment is considered generally safe, a recent systematic review concluded that BoNTA injection could results in mandibular bony change, which may hinder BoNT-A benefits [17].